# Exercise Prescription Principles among Physicians and Physical Therapists for Patients with Impaired Glucose Control: A Cross-Sectional Study

**DOI:** 10.3390/jfmk8030112

**Published:** 2023-08-07

**Authors:** Michael A. Petrie, Kristin A. Johnson, Olga Dubey, Richard K. Shields

**Affiliations:** Department of Physical Therapy and Rehabilitation Science, Carver College of Medicine, The University of Iowa, Iowa City, IA 52242, USA; michael-petrie@uiowa.edu (M.A.P.); kristin-a-johnson@uiowa.edu (K.A.J.); olga-dubey@uiowa.edu (O.D.)

**Keywords:** exercise, recommendation, diabetes, glucose, clinical practice

## Abstract

Exercise confers a multitude of benefits with limited adverse side effects, making it a powerful “medication” for a plethora of diseases. In people living with uncontrolled glucose levels, exercise can be an effective “medication” to assist in the management of hyperglycemia. We sought to survey healthcare providers (physicians and physical therapists) to determine the current state of exercise recommendation for people with glucose control issues. Healthcare providers were surveyed from six academic medical centers in the Midwest to determine the recommended exercise parameters (type, frequency, duration, intensity, and timing) for patients with glucose control issues. Data from 209 practitioners who completed the survey were used for analysis. Chi-square tests were used to determine differences in exercise recommendations between physical therapists (PTs) and physicians (MD/DOs). PTs and MD/DOs recommended similar exercise parameters. Of all respondents, 78.9% recommended exercise to patients with glucose control issues. Respondents who considered themselves to be active exercisers were more likely to recommend exercise than those who were not exercisers. Only 6.1% of all respondents recommended post-meal exercise. Healthcare providers overwhelmingly recommended exercise for people with glucose control issues, but the “timing” is not congruent with best practice recommendations.

## 1. Introduction

Americans are becoming less physically active [1], propelling an epidemic of non-communicable diseases [2,3,4], including diabetes [4]. Nearly half of the US population lives with prediabetes (38.0%) or diabetes (11.3%) [5], a concerning percentage given this metabolic disease is a primary predictor of all-cause mortality [6]. Type II Diabetes is characterized by uncontrolled blood glucose levels that develop when the body loses its sensitivity to insulin [7]. Therefore, medications that assist in regulating blood glucose are often the primary prescription for people with this disease. In the United States, the annual cost of insulin and anti-diabetic medications exceed $30 billion dollars and comprises one of the largest healthcare expenditures for the disease [8]. Exercise, often referred to as “medicine”, is another intervention that assists in regulating blood glucose and insulin levels in people with diabetes [9], is cost-effective [10], and promotes a “patient-centered” lifestyle approach to metabolic health [11].

Exercise improves metabolism by the activation of skeletal muscle [12,13,14]. Skeletal muscle is a conduit to biomechanical performance but is also a powerful endocrine organ [15] that, when activated, secretes hormones (myokines) [13] that reduce systemic inflammation, improve metabolism, and regulate blood glucose levels in people with and without glucose intolerance [9,16]. Skeletal muscle has a strong effect on blood glucose in that it uses more than 80% of the substrate during exercise if the timing is congruent with a meal [17,18]. Glucose enters skeletal muscle from the bloodstream via insulin-dependent and insulin-independent pathways [19,20]. In people living with diabetes, the insulin-dependent pathways are impaired, yet the insulin-independent pathways, like the AMP-activated protein kinase (AMPK) pathway, remain intact [21,22]. Muscle contraction triggers the AMPK pathway to mobilize the GLUT4 glucose transport protein to skeletal muscle cell surfaces to open the “door” for glucose to enter the cell [23]. Importantly, activity and exercise reduce blood glucose levels and decreases peak insulin following a meal [16]. Chronic exposure to elevated levels of insulin after a meal may ultimately contribute to reduced insulin sensitivity [24,25]. While the long-term adaptations to insulin receptors are well understood through chronic exercise studies, acute responses of exercise on insulin and glucose, after a meal, are now contributing to recommendations for “movement or exercise” after eating as a lifestyle behavior [26]. Accordingly, a lifestyle of routine muscle activation by bouts of exercise mitigates the development of diabetes through chronic adaptations [4] but also via acute reduction of needed insulin after a meal [16].

The recommended “dose” of exercise for people living with diabetes was recently updated by the American College of Sports Medicine (ACSM) [26] and the American Diabetes Association (ADA) [27] to include post-prandial exercise [16]. Many of the published exercise guidelines are similar for people with and without metabolic disorders, including the recommendation to accumulate a minimum of 150 min/week of moderate-to-vigorous aerobic exercise spread over at least three days. Resistance exercise is also recommended at least two days per week, and most guidelines recommend combining aerobic and resistance training within the same session [28]. The exercise guidelines specific for people living with diabetes also recommend exercise after eating, or post-prandial exercise, as it attenuates the hyperglycemic and hyper-insulinemic loads from a meal [16]. Various types of exercise are effective for mitigating the post-prandial glycemic response [29] and greater benefits occur when exercise is performed promptly after eating [30]. In light of the recently updated exercise guidelines, we sought to determine if the exercise recommendations provided by healthcare practitioners for people with glucose control issues include post-prandial exercise. 

The purpose of this study was to determine the congruency of the recommended dose of exercise (type, frequency, duration, intensity, timing) for people who have glucose control issues among practicing physicians and physical therapists. Our primary interest related to whether healthcare practitioners include post-prandial exercise as part of a routine prescription. We suspected that the timing of exercise relative to meals is not typically prescribed by practitioners who treat people with glucose control issues. 

## 2. Materials and Methods

### 2.1. Study Design, Development, and Setting

Using a secure online survey platform (Qualtrics), a 10-question survey was developed to perform a cross-sectional study of healthcare practitioners regarding their recommended exercise parameters for people with glucose control issues between June 2022 and January 2023. The survey ascertained the exercise prescription practice and exercise status of the physicians and physical therapists, determined the healthcare providers’ exercise- based referral practices, and ascertained the recommended “dose” of exercise for people with glucose control issues by determining the type, frequency, duration, intensity, and timing of their recommended exercise prescription. The survey was piloted among a team of physicians and physical therapists (*n* = 10) to establish face validity. The survey was designed to take less than 3 min as recommended by the survey review team. This survey and research protocol was approved by the University of Iowa Institutional Review Board (IRB). The final research protocol was congruent with the planned protocol as approved by the IRB.

### 2.2. Survey Participants

Eligible participants included English-speaking physicians (MD/DO) and physical therapists (PT) with email accounts across six major academic medical institutions in the Midwest of the United States. An email invitation to participate in the survey was delivered to 10,059 healthcare professionals (9411 MD/DO, 648 PT). Survey questions were reviewed and modified based on input from a separate panel of physical therapists and physicians who had expertise in treating metabolic disease. As part of that review, we included a question related to the personal active exercise habits of the respondents. We also focused the survey so that only participants who treated people with glucose control issues were included in the final analysis. An example of the survey is available in Appendix A. 

### 2.3. Statistical Analysis

Cronbach’s Alpha was calculated to assess the internal consistency (reliability) of the survey. A percentage of physician and physical therapist responses was calculated for each survey question. Chi-square tests of homogeneity were used to evaluate for differences in exercise recommendation percentages among and between physicians and physical therapists. Chi-square goodness-of-fit tests were used to evaluate for differences in response from an expected probability frequency estimated for each question. The expected frequency (null hypothesis) was based upon the number of answer options per question (e.g., a question with four answer options had an expected frequency of 25%). Significance was set at *p* < 0.05.

## 3. Results

### 3.1. Survey Properties, Exercise Advice, and Exercise Status

Among the 1649 practitioners who opened the email (1542 MD/DO, 107 PT), 216 practitioners started the survey. Data from the 209 practitioners who completed the survey (166 MD/DO, 43 PT) were used for data analysis. The average time for respondents to complete the survey was 1 min and 40 s.

Cronbach’s Alpha was 0.81, supporting a high internal consistency (reliability) of the survey. One hundred percent of the pre-study expert panel (5 physical therapists; 5 physicians) indicated that the survey captured key principles necessary to understand exercise prescription for patients with glucose control issues. 

Similar proportions of physical therapists and physicians offer exercise-related advice for people with glucose control issues, with 78.9% of all surveyed practitioners indicating they advise exercise (Figure 1A, *p* = 0.216). The proportion of practitioners who considered themselves to be active exercisers was similar between physical therapists and physicians, and 87.1% of practitioners reported that they actively exercise (Figure 1B, *p* = 0.428). There were similar groupings of practitioners by exercise-advice status and personal exercise status between physical therapists and physicians (Figure 1C, *p* = 0.345). When practitioners were grouped by their personal exercise status, active exercisers were 31.1% more likely to offer exercise-related advice to patients with glucose control issues than practitioners who were non-exercisers (Figure 1D, *p* < 0.001).

### 3.2. Exercise Type, Frequency, Duration, and Intensity

Most of the practitioners who advise patients about exercise recommend more than one type of exercise: 17.6% recommend one type, 30.9% recommend two types, 50.3% recommend three types, and 1.2% recommend four types of exercise. As compared to physicians, physical therapists were 24.4% more likely to recommend anaerobic exercise (*p* = 0.013), but the professions were similar in their recommendations of other types of exercise (all *p*’s > 0.193), with 91.5% of all practitioners recommending aerobic exercise and 82.4% recommending general physical activity (Figure 2A). Physical therapists and physicians recommend different exercise frequencies (*p* < 0.001), with physical therapists more likely to prescribe >3 but ≤5 sessions and physicians more likely not to specify the number of sessions (Figure 2B). Together, 6.7% and 25.5% of practitioners recommend >1 but ≤3 sessions and >5 sessions, respectively. Physical therapists and physicians recommend similar exercise durations (*p* = 0.256), and they most commonly recommend >20–30 min per session (Figure 2C). Recommended exercise intensities were similar between physical therapists and physicians (*p* = 0.076), with the majority of practitioners not specifying exercise intensity based on maximal heart rate (Figure 2D).

### 3.3. Exercise Timing

Physical therapists and physicians recommend similar timing of exercise (*p* = 0.432), with most practitioners not recommending a specific time (57.0%), or they recommend anytime (40.0%) (Figure 3A). Physical therapists and physicians were similar in recommending exercise timed to eating (*p* = 0.092), with the vast majority of practitioners not recommending exercise based on the time of a meal (Figure 3B).

### 3.4. Exercise Referrals

Physical therapists and physicians differed in referral practices (*p* = 0.029) as physical therapists were 9.2% more likely than physicians to “always” refer patients with glucose control issues to exercise specialists (Figure 4A). Over 50% of all practitioners indicated they “sometimes” refer to specialists. Among the referring practitioners, 47.4% refer to one specialty, 42.4% refer to two specialties, and 10.2% refer to three specialties. The types of exercise specialists to whom practitioners refer were similar between physical therapists and physicians (Figure 4B, *p* = 0.209), with physical therapists being the specialist of choice for both physicians and physical therapists (76.3%).

## 4. Discussion

The primary purpose of this study was to determine the congruency of healthcare practitioners’ exercise recommendations for people with glucose control issues. The survey revealed several key findings: 1. most practitioners offer exercise-related advice; 2. practitioners who are active exercisers are more likely to prescribe exercise; 3. physical therapists and physicians prescribe similar exercise protocols in accordance to guidelines, and 4. physical therapists and physicians do not prescribe post-prandial exercise which is not congruent with current exercise guidelines. The high prevalence of offering exercise-related advice from these practitioners indicates healthcare providers are generally aware of the benefits of exercise for people with metabolic impairment, but inconsistencies in the timing of exercise after a meal suggest a general lack of specific knowledge about current “timing” recommendations for people with blood glucose control issues.

### 4.1. Exercise Prescription Practices Compared to Recommended Exercise Guidelines

The first major revelation from this survey is that only 6.1% of healthcare practitioners recommend post-prandial exercise for people with glucose control issues. This finding is not congruent with the updated ACSM guidelines [26] and suggests that healthcare practitioners, both physicians and physical therapists, are not aware of the contemporary research supporting exercise after a meal to reduce post-prandial insulin and glucose in people with metabolic disorders [16]. Various types of exercise such as walking, cycling, and resistance exercise are effective for reducing post-prandial glucose levels [29]. Initiating exercise promptly after a meal is important as the post-prandial hyperglycemic peak is blunted more by exercise that begins <30 min after eating as opposed to starting ≥30 min [29,30]. Post-prandial exercise not only reduces blood glucose levels but, perhaps more importantly, is associated with lower amounts of insulin in the bloodstream [31,32], thereby reducing a factor that precipitates insulin receptor desensitization and perhaps the development of diabetes [24,25].

Despite the lack of congruency with post-prandial exercise recommendations for both physicians and physical therapists, nearly 80% of all practitioners offer exercise-related advice to people with glucose control issues. This finding is consistent with the high prevalence of primary care physicians (93.3%) who “often” or “always” prescribe physical activity for adult patients with chronic disease [33] and with prescription practices from 2010 in which 76% of physicians reported providing weight-loss counseling and physical activity instruction for patients with diabetes [34]. However, our findings confirmed that both physical therapists and physicians who consider themselves active exercisers were 31.1% more likely to offer exercise-related advice when compared to their non-exercising counterparts. This finding is consistent with exercise prescription patterns for people with cardiovascular disease [35].

When types of exercises were considered, among both physical therapists and physicians, aerobic exercise was the most recommended type of exercise (91.5%), followed by general physical activity (82.4%). These recommendations are congruent with ACSM exercise guidelines that recommend adults with diabetes participate in regular aerobic exercise and that “some physical activity is better than none” [26]. Anaerobic exercise was not prescribed as often as the other types of exercise (57.6%) among both groups of practitioners. Current guidelines recommend at least 2 days of resistance training per week as it improves several metabolic biomarkers, increases insulin sensitivity, and reduces A1C [26]. Practitioners did not recommend a specific time of day to exercise (57.0%), or they recommended exercising at any specific time of day (40.0%). There is some support for better glycemic control [36] and insulin sensitivity [37] in people with metabolic disorder when exercise is performed in the afternoon as opposed to the morning, but this work is still evolving as we discover more about circadian rhythms and metabolic gene regulation [38,39]. Current ACSM guidelines do not recommend a specific time of day to exercise, citing a need for more research in this area [26].

### 4.2. Similar Exercise Prescription Practices between Physical Therapists and Physicians

There was internal consistency between physical therapists and physicians among several aspects of exercise prescription. Physical therapists and physicians recommend similar times of day to exercise, timing of exercise to meals, durations of exercise, and exercise intensity based on maximal heart rate. Both professions refer to similar types of exercise specialists. Physicians and physical therapists differed in exercise frequency recommendations, with physical therapists more likely to recommend >3 but ≤5 sessions per week and physicians more likely not to specify exercise frequency, a finding that is not congruent with best practice. Physical therapists were also more likely than physicians to recommend anaerobic exercise and to “always” refer patients with glucose control issues to another physical exercise specialist. Increasing referrals to exercise specialists may improve exercise recommendations by overcoming some of the barriers to exercise prescription [40].

### 4.3. Limitations

This survey does not elucidate the specific exercise prescriptions of healthcare providers but focuses on key concepts or principles that form the framework for sound exercise prescription. Our data were limited in that they represented a broad swath of healthcare providers across the Midwest, so we cannot ascertain if prescriptions vary based on geographical locations. Although we narrowed the survey so that the healthcare providers made recommendations only for patients with “blood glucose control” issues, we did not hone in on the specific diagnoses. Thus, we cannot link these recommendations to any specific patient type. Lastly, our sample size did not enable us to analyze stratified data based on characteristics of the healthcare centers for which the respondents were employed and the personal exercise habits of all practitioners. Future investigations are warranted to better understand if respondents’ variations in practice influenced their prescriptions for exercise. Nonetheless, this study offers an important finding, not previously discovered, regarding post-prandial exercise prescription among physicians and physical therapists that is not congruent with contemporary guidelines for people with glucose control issues. 

## 5. Conclusions

In summary, physicians and physical therapists are congruent in their exercise prescription principles for people with glucose control issues. Importantly, both physicians and physical therapists are not congruent with “newer” recommendations related to post-prandial “timing” of exercise among people with glucose control issues. Future work is underway in our lab to better understand the precise dose of post-prandial exercise to optimize insulin response in people with and without disability.

## Figures and Tables

**Figure 1 jfmk-08-00112-f001:**
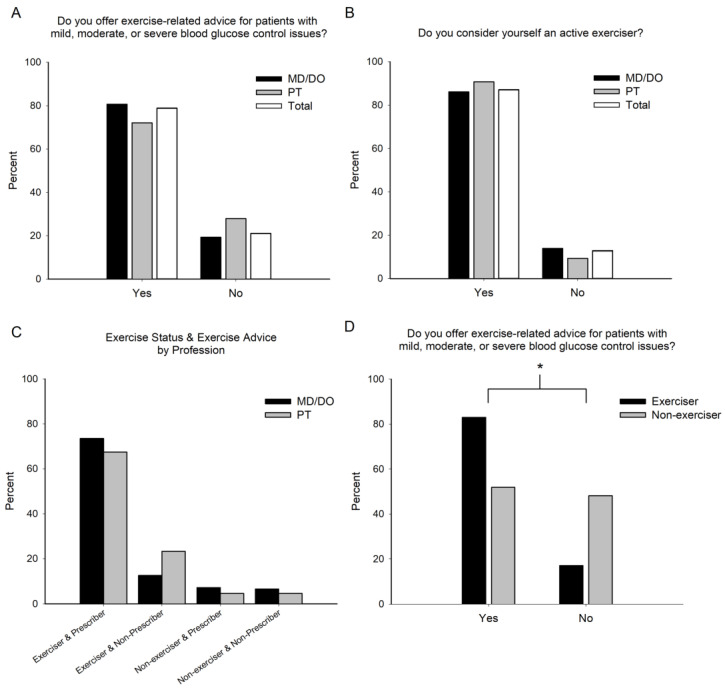
Exercise Advice and Exercise Status. The percentage of surveyed healthcare practitioners who offer exercise-related advice (**A**) and those who consider themselves to be active exercisers (**B**). The percentage of healthcare providers grouped by their exercise advice and personal exercise status (**C**). Exerciser and prescriber: those who recommend exercise for people with glucose control issues and who consider themselves to be active exercisers. Exerciser and non-prescriber: those who do not recommend exercise for people with glucose control issues and who consider themselves to be active exercisers. Non-exerciser and prescriber: those who recommend exercise for people with glucose control issues and who do not consider themselves to be active exercisers. Non-exerciser and non-prescriber: those who do not recommend exercise for people with glucose control issues and who do not consider themselves to be active exercisers. The percentage of healthcare providers who offer exercise-related advice for people with glucose control issues based on their personal exercise status (**D**). Physicians (MD/DO); physical therapists (PT); combination of the two (Total). * Indicates a significant (*p* < 0.05) difference between healthcare providers who consider themselves to be active exercisers and those who do not consider themselves to be active exercisers.

**Figure 2 jfmk-08-00112-f002:**
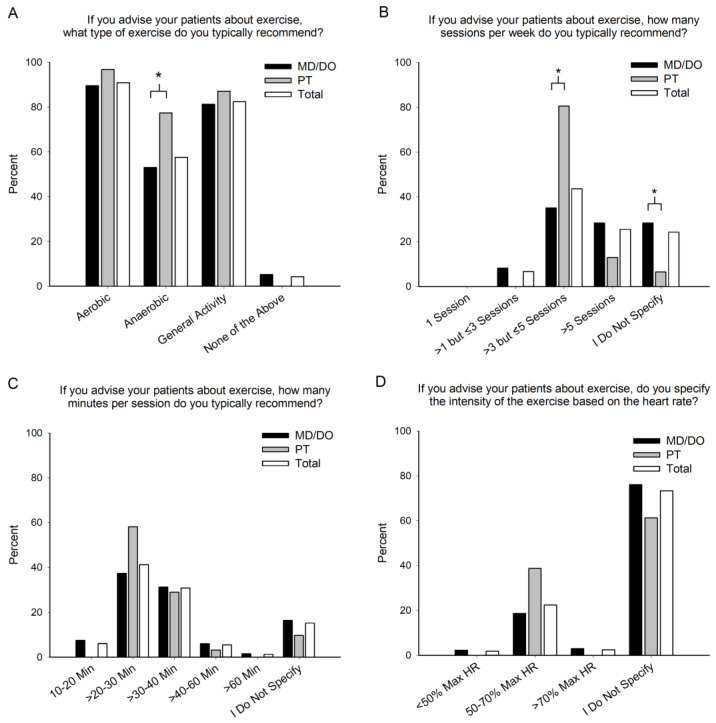
Exercise Type, Frequency, Duration, and Intensity. The type (**A**), frequency (**B**), duration (**C**), and intensity (**D**) of exercise recommended by healthcare providers for patients with glucose control issues. Physicians (MD/DO), physical therapists (PT), and combination of the two (Total). * Indicates a significant (*p* < 0.05) difference between MD/DO and PT.

**Figure 3 jfmk-08-00112-f003:**
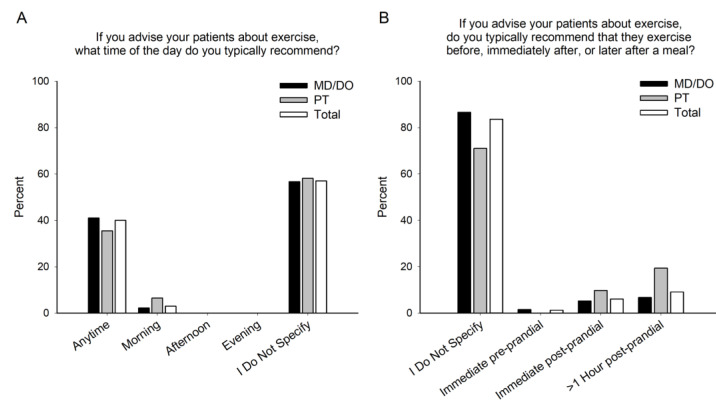
Exercise Timing. The time of day to exercise (**A**) and the timing of exercise relative to meals (**B**) recommended by healthcare providers who offer exercise-related advice for patients with glucose control issues. Physicians (MD/DO), physical therapists (PT), and combination of the two (Total).

**Figure 4 jfmk-08-00112-f004:**
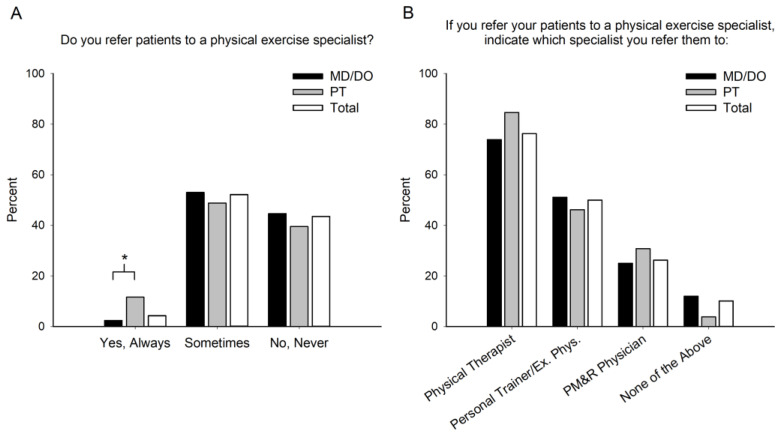
Exercise Referrals. Healthcare providers’ referral practices (**A**) and exercise specialists to whom they refer patients with glucose control issues (**B**). Physicians (MD/DO), physical therapists (PT), and the combination of the two (Total). * Indicates a significant (*p* < 0.05) difference between MD/DO and PT.

## Data Availability

Data is available upon requests made to the corresponding author.

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
