# Peer review of "Exercise Prescription Principles among Physicians and Physical Therapists for Patients with Impaired Glucose Control: A Cross-Sectional Study"

_jfmk, 2023, doi:10.3390/jfmk8030112_

Round 1
Reviewer 1 Report
The authors have submitted the paper “Exercise prescription principles among physicians and physical therapists for patients with impaired glucose control” with the stated study purpose of “to determine the congruency of the recommended dose of exercise (type, frequency, duration, intensity, timing) for people who have glucose control issues among practicing physicians and physical therapists.” After reviewing, it is requested that the following concerns be addressed:
Major Concerns:
Reporting
i) This cross sectional study design aligns with and should be reported using the Equator Network STROBE reporting guidelines for a cross-sectional study design. The authors should complete this checklist and report along these lines.
Methods:
i) The authors state the study was approved by the University of Iowa Institutional Review Board. There is no indication if the study was registered on the Open Science Framework or comment on if the final process differed to what was planned. Please include this to allow for appropriate evaluation.
ii) In its reported format, this study is not able to be replicated. There is a lack of detail on the survey design, the processes used to determine survey validity and reliability and the recruitment methods. Further information needs to be reported to demonstrate the steps taken to determine if the survey instrument provided the information required to meet the study purpose.
iii) The Survey Distribution sub-heading (lines 92-100) is reporting results and not methods. This should be moved to the results section.
iv) Statistical Analysis requires further detail to describe all analysis undertaken (frequencies reported as percentages etc).
Discussion:
i) 4.1 – need to provide consistent context of results in relation to recommendations. Does the advice given meet the current recommendations? It is not clear if all advice provided by respondents meets these standards.
ii) 4.2 – need to provide context of who is providing the advice based on guidelines. It is reported that both are providing similar advice for times of day to exercise, timing of exercise to meals, duration of exercise and exercise intensity, but there is no reporting of if this is the best advice and consistent with guidelines. Similarly, when advice is reported to be different between disciplines, there is no mention of which advice is closest to the current recommendations.
iii) 4.3 – The reviewer is not clear as to the purpose of this section. There is no previous mention of barriers to exercise prescription or indication if the survey asked questions to this end. There is no direct link to the results. Please remove or report results around barriers if this was collected from the survey.
Minor Concerns:
Introduction
i) The background information is well considered, however, there is a lack of information on the link between exercise advice and the stated purpose.
Conclusion
Lines 297-301 present new information that is not discussed in the paper. The conclusion should only summarise that which has already been presented and not introduce new ideas. Please revise the conclusion.
Survey
Good practice would have the survey (or at least the exact survey questions) included as an appendix to allow for replication and further investigation. Please consider including this.
Reviewer 2 Report
The article is skillfully organized and addresses a highly relevant topic, highlighting its significance in establishing future guidelines for healthcare professionals involved in the prevention and management of type 2 diabetes. The added relevance of the paper is to focus on the importance of the timing of exercise in order to reduce post-prandial insulin and glycemic peak and, more in general, to have a better blood glucose levels control in subject predisposed to diabetes.
The sampling methodology presents some weakness:
1) The survey distribution methodology used resulted in a very low response rate. Out of the 10,059 emails that were delivered, only 16% among MD/DO and PT opened the email, and a mere 2% of them completed the survey. Despite the ease and affordability of this type of distribution, did the authors consider alternative options? Please address limitations and strengths of your choice.
2) Within the study's limitations, the authors acknowledged that their sample does not represent the geographical area under investigation. Do the authors consider that another potential bias could be that practitioners who are more aware of the benefits of exercise were more likely to respond?
